# The Effect of Graphene Nanofiller on the Surface Structure and Performance of Epoxy Resin–Polyhedral Oligomeric Silsesquioxane (EP-POSS)

**DOI:** 10.3390/nano11040841

**Published:** 2021-03-25

**Authors:** Yanhong Fang, Ping Wang, Lifang Sun, Linhong Wang

**Affiliations:** 1State Key Laboratory of Power Transmission Equipment & System Security and New Technology, Chongqing University, Chongqing 400044, China; 20181101005@cqu.edu.cn; 2School of Electrical Engineering, Chongqing University, Chongqing 400044, China; 3Department of Pharmacy, Sichuan Vocational College of Health and Rehabilitation, Zigong 643000, China; sunlifang0524@126.com; 4School of Smart Health, Chongqing College of Electronic Engineering, Chongqing 400013, China; 201018044@cqcet.edu.cn

**Keywords:** epoxy resin, polyhedral oligomeric silsesquioxane, nanoparticle, hydrophobicity, graphene

## Abstract

Epoxy resin–polyhedral oligomeric silsesquioxane (EP-POSS) has excellent mechanical properties and hydrophobic properties. In order to adapt for application in sensor and photovoltaic fields, graphene, nano-SiO_2_ and nano-ZnO were used to modify EP-POSS. FTIR was used to characterize changes on the surface structure after introducing nanoparticles. The change of hydrophobicity was measured using a contact angle test. TEM test results showed that nanoparticles were successfully inserted between the graphene sheets. However, the content of Si on the surface was low, as the cage structure of POSS in the molecular chain was coated by epoxy groups. XRD tests indicated that nanoparticles facilitated the dispersion of graphene in EP-POSS. XPS characterized the chemical state and content of the elements, confirming that the addition of graphene can induce the enrichment of Si on the surface of EP-POSS, which had a shielding effect on the main chain and improved the hydrophobicity. Wear resistance and adhesion tests showed that, after the introduction of nanoparticles, the EP-POSS coating film met the requirements of graphene materials.

## 1. Introduction

Epoxy resin-polyhedral oligomeric silsesquioxane (EP-POSS) can be used in the fields of sensors and photovoltaics, and it is of great interest to the industry because of its effectiveness, economy, and ease of use [1,2,3]. However, some defects (pores, etc.) in EP-POSS materials may cause their use to be limited. Graphene is a carbon material composed of a single-atom structure and has photosensitive properties. In addition, graphene with its two-dimensional structure has excellent physical and chemical properties and has been widely used in optical, sensor, and other fields [4,5]. Therefore, it is necessary to add a certain amount of graphene, nano-SiO_2_, and nano-ZnO to modify the surface structure and properties of EP-POSS.

POSS is characterized by a regular structure, small scale (about 1 nm), good compatibility, and excellent performance [6,7]. It can form true nanoscale and molecular hybrid materials. The general formula for the chemical structure of sesquioxane is [(RSiO1.5) N], where N is 6, 8, 10, etc., which can be written as T6, T8, T10, etc. One of the most studied compounds is the symmetric T8 compound. It is mainly composed of two parts: (i) The cage structure core of the inorganic hexahedron is composed of a Si–O–Si bond, which makes the molecule have good thermal stability. (ii) The shell formed by organic functional groups is connected to the eight vertices of a Si atom, where the R group can be an active or inert group, including a hydrogen group, alkyl group, enyl group, aryl group and their derivative groups [8,9].

Some important discoveries have been made in the study of modified POSS materials [10,11,12,13,14,15,16,17,18,19,20]. For example, Ye et al. [10,11] synthesized tetraaniline polyhedron oligosiloxane molecules (TA-POSS) and prepared functionalized molecules (POSS-TA-G) using graphene (G) as a dispersant. The functional epoxy composite coating was prepared by coating Q235 steel with POSS-TA-G. The effects of POSS-TA-G on the barrier property and self-healing property of the coating were investigated using electrochemical impedance spectroscopy (EIS) and local electrochemical impedance spectroscopy (LEIS). It was found that when 0.5 wt% POSS-TA-G was added, the coating had good corrosion resistance. Graphene could improve its effective shielding performance and self-healing ability brought by POSS-TA. Rodowek et al. [14,15] studied the composite effect of nano polyhedral sesquioxane modified polydimethylsiloxane-based protective coatings. In situ vibration spectra of the coating were determined to track the band variation of the protective coating during forced anodic polarization. Li et al. [19,20] obtained modified cyanoate resin (G-CE) and functionalized silicon fiber (K-SFS) by introducing glycidyl ester polyheoheplosesquisiloxane (G-POSS) and a γ-aminopropyl triethoxy silane coupling agent (KH-550). K-SFS/G-Ce transparent composites were prepared from G-Ce and K-SFS. It was found that the interface compatibility of K-SFS/G-Ce transparent composites was enhanced, and that the mechanical properties of the materials ere improved. Joseph D. Lichtenhan expounded that the combination of a functional polymer and inorganic nanometer compound structure has become a new research direction, and that the development and new types of monomers and polymers based on inorganic polyhedrons (POSS system as described here) can be used to strengthen the development of a wide range of new materials for materials chemists, scientists and engineers have potential uses [21,22]. The abovementioned studies have laid a solid research foundation for the development of modified POSS materials. However, the surface structure and properties of modified EP-POSS materials are somewhat lacking.

In this paper, based on the shortcomings of the studies above, EP-POSS was modified using graphene, nano-SiO_2_, and nano-ZnO, and the surface structure, hydrophobicity, microstructure, and crystal phase characteristics of the modified EP-POSS materials were studied by infrared spectroscopy (FTIR), contact angle, TEM, and XPS, respectively. Secondly, the physical properties of modified EP-POSS were verified using a wear resistance test and adhesion test.

The rest of this article is divided into three parts. In the first part, the materials and test methods used for the modified EP-POSS are introduced. In the second part, the surface structure and properties of the modified EP-POSS are analyzed in detail. Finally, in the final section the research results are summarized.

## 2. Experiments

### 2.1. Raw Materials

The EP-POSS resin was home-made in the laboratory; it was prepared by a condensation reaction with isopropanol (IPA), 5% tetramethylammonium hydroxide (TMAH) aqueous solution and γ-(2,3-glycidoxy) propyltrimethoxysilane (KH560), the specific preparation method is referred to in previous research [23]. The molecular structure of POSS is shown in Figure 1.

SiO_2_ nanoparticles with a particle size of 20 nm were purchased from Macklin Reagent Company (Shanghai, China); ZnO nanoparticles were a 30wt% dispersion with a particle size of 30 nm, and were purchased from Macklin Reagent Company (Shanghai, China); and the thickness of graphene was 0.2 nm, the size was 2 μm, and the specific surface area was 800 mm^2^/g, the graphene was purchased from Macklin Reagent Company (Shanghai, China). All substances were used without further processing after purchase.

### 2.2. Sample Preparation

EP-POSS/graphene was prepared by adding 6 wt% graphene to EP-POSS. The specific preparation method was as follows: 8.0 g EP-POSS and 0.5 g graphene were added into a centrifuge tube, stirred evenly, and dispersed ultrasonically for 30 min to obtain EP-POSS/graphene.

EP-POSS/graphene/SiO_2_ was prepared by adding 6 wt% graphene and 6 wt% SiO_2_ to EP-POSS. The specific preparation method was as follows: 8.0 g EP-POSS, 0.5 g graphene, and 0.5 g nano-SiO_2_ were successively added into a centrifugation tube, and then the EP-POSS/graphene/SiO_2_ was obtained by ultrasonic dispersion after stirring evenly for 30 min.

EP-POSS/graphene/ZnO was prepared by adding 6 wt% graphene and 6 wt% ZnO to EP-POSS. The specific preparation method was as follows: 8.0 g EP-POSS, 0.5 g graphene, and 1.67 g nano-ZnO dispersions were successively added into a centrifugation tube, and then the dispersions were stirred evenly and dispersed ultrasonically for 30 min to obtain EP-POSS/graphene/ZnO.

Glass slides were coated with EP-POSS nanocomposite and dried at 150 °C for 15 min to obtain EP-POSS nanocomposite films with a thickness of about 30 μm.

### 2.3. Test Method

Before the measurements, the EP-POSS sample (0.1–0.2 g) was degassed in a vacuum at 110 °C for 12 h to remove impurities. The sample contact angle test data were recorded using an OCA 100 automatic video optical contact angle instrument from Dataphysics (Beijing, China). A Bruker VERTEX 80v spectrometer (Bruke, Germany), with a test wavelength of 400–4000 cm^−1^ and resolution of 2 cm^−1^ was used to test the tablet. XPS analyses used a Thermo Scientific ESCALAB 250Xi instrument (ThermoFisher SCIENTIFIC, Wuhan, Hubei Province, China), an aluminum X-ray source, and a working power of 200 W. The vacuum pressure range was 10^−9^–10^−8^ Pa, and Thermo Scientific Avantage 4.52 software was used for data collection and processing. A DX2700 X-ray diffractometer (XRD) (Haoyuan instrument, Dandong, Liaoning Province, China) was used for characterization. The accelerating voltage was 40 kV, the current was 30 mA, the step angle was 0.02°, and the scanning range was 5–90°. The TEM test was performed using a JEM-2100F Japan Electronics Corporation to observe and analyzed the nanostructure characteristics of EP-POSS modified by nanoparticles. Referring to GB/T 5210 (《GB 5210-1985 Determination of adhesion of coatings—Pull method》, This standard is applicable to the quantitative Determination of adhesion between monolayer or composite coatings and substrates or between coatings.) for the testing procedure, an Instron 23.8 million capacity testing machine was used, and the rate of increase in tension was 1 MPa/s. Referring to EN 6602 for the testing procedure, an RCA abrasion tester (Norman tool, Evansville, IN, USA) was used, and the rotating friction was measured at over 5000 revolutions to test the volume loss.

## 3. Results and Discussion

In order to study the changes of the surface structure of EP-POSS after the addition of nanoparticles, FTIR was used for structural analyses, and the infrared spectrum is shown in Figure 2. By the spectrum of A can be found in Figure 2, after joining the graphene and nanoparticles, we found a weak peak at 1634 cm^−1^, which was the hydroxyl stretching vibration peak and H_2_O variable angle vibration peak. At the same time, the wide peak at 3442 cm^−1^ was the -OH stretching vibration peak and showed that a certain amount of moisture was found in the resin system, this was due to the large specific surface area of the graphene and nanoparticles. The C-H stretching vibration peaks at 2918 cm^−1^ and 2852 cm^−1^ and the C-O stretching vibration at 1201 cm^−1^ were enhanced after graphene was added to EP-POSS, which was due to the lattice defects at the edges of the graphene and the existence of a number of C-H and C-O bond structures. The peaks at 1105 cm^−1^ and 1040 cm^−1^ were Si-O-Si antisymmetric stretching vibration peaks and C-H in-plane bending vibration peaks, respectively. The peaks became dull after graphene was added, indicating that graphene and nanoparticles formed a complex with EP-POSS. The strength of the stretching vibration peak of Si-O at 788 cm^−1^ was slightly increased compared with the nearby peak of 742 cm^−1^, indicating that the addition of graphene with large specific surface area could weaken the shielding effect of the epoxy chain segment on the POSS group, resulting in the characteristic absorption peak of the stretching vibration of Si-O being slightly enhanced. SiO_2_ and ZnO inorganic nanoparticles were added to EP-POSS, and their infrared spectra did not change significantly compared with that of EP-POSS with only graphene added. Other test methods are explained below to characterize the impact of the inorganic nanoparticles.

The introduction of nano-SiO_2_ can increase the content of Si element in EP-POSS, and nano-ZnO can improve the antibacterial effect of EP-POSS. so how does the introduction of nanoparticles affect the surface of EP-POSS? A contact angle test was used for characterization, and the test results are shown in Figure 3. Figure 3 shows that after adding different nanoparticles, the difference in the contact angle of EP-POSS was contrary to expectations. According to previous research work [24], the average contact angle of pure EP-POSS is 101°. After graphene is added to EP-POSS, the contact angle is 100.1°, which is lower than that of EP-POSS. However, after SiO_2_ was added, the contact angle was significantly reduced to 58.9°, which showed that the introduction of SiO_2_ caused a sharp drop in hydrophobic properties compared to graphene. After ZnO/graphene was added, the contact angle was lower than that of SiO_2_/graphene and decreased to 44.5°, which indicated that the additional introduction of inorganic nanoparticles negatively affected the surface hydrophobicity.

The contact angle analyses showed that the hydrophobicity of the EP-POSS surface was substantially reduced when nano-SiO_2_ and ZnO were added compared to graphene. To study the effect of the addition of inorganic nanoparticles on the surface morphology, the prepared film was tested using TEM. Figure 4 shows TEM images of EP-POSS and samples with different nanofillers. In Figure 4a, EP-POSS is a sphere with a diameter of 200 nm under a TEM electron microscope. This is because the POSS group is coated by epoxy segments and POSS can easily crystallize to form a compact spherical structure. In Figure 4d, the spherical structure has disappeared after adding graphene to EP-POSS, indicating that the crystalline structure of POSS was destroyed and the coating effect of the epoxy segment on the POSS group was weakened. At the same time, the color of the TEM image is darker because the graphene is unevenly dispersed, and the layer structure of the graphene is in a stacked state. Figure 4b,c shows EP-POSS images with nano-SiO_2_ and nano-ZnO added, respectively. It can be seen that there were SiO_2_ and ZnO nanoparticles between the graphene sheet structures, and the TEM image is obviously lighter than that in Figure 4d. This is because the SiO_2_ and ZnO nanoparticles enter the gap between the graphene sheets and the graphene spreads out and the dispersion is more uniform. These test results show that the introduction of graphene improves the surface hydrophobicity compared to inorganic nanoparticles, and that the surface inertness and two-dimensional structure of graphene are the main sources of this effect. However, the addition of inorganic nanoparticles can improve the compatibility of graphene and EP-POSS.

Both POSS groups and nanoparticles in EP-POSS have a crystal structure. X-rays are diffracted when they pass through the crystal structure. The orientation and intensity of the diffraction lines in space are related to the crystal structure, so XRD is used to characterize the nanoparticles. The crystal phase structure of EP-POSS test results are shown in Figure 5. Figure 5 shows that after nanoparticles are added to EP-POSS, there are only 20.9° POSS diffraction peaks and 26.3° graphene (002) crystal plane diffraction peaks, but no 24.1° SiO_2_ diffraction peaks and 34.4° diffraction peaks are found. The diffraction peak of ZnO indicates that nano-SiO_2_ and ZnO have good dispersibility in EP-POSS. In Figure 5 EP-POSS/Gr, the 26.3° graphene has the strongest crystal plane diffraction peak because graphene has poor compatibility with EP-POSS, so it is easy to stack on the surface, and the peak intensity is larger. After nano-SiO_2_ and ZnO were added, the characteristic peak of graphene weakened, which indicated that the addition of nano-SiO_2_ and ZnO helped graphene disperse in EP-POSS. The reason for this is that nano-SiO_2_ and ZnO are inserted into the graphene sheet, which reduces the van der Waals force and destroys the crystal phase structure between graphenes, so the characteristic peaks of graphene are significantly weakened. In addition, the EP-POSS molecular chain grows in the gap between the nanoparticles, and the compatibility between the POSS structure and the nanoparticles increases the compatibility of graphene in EP-POSS, which is in the microstructure shown using TEM. The compatibility mechanism is shown in Figure 6. The diffraction peak of POSS at c is strongest at 20.9°, which indicates that the addition of graphene reduced the shielding effect of the epoxy molecular chain in EP-POSS on the POSS group.

XPS is a surface analysis method that uses X-ray radiation samples to measure the energy and quantity of photoelectrons to determine the binding energy of electrons to identify the chemical properties and composition of a sample’s surface. It can test the surface chemical information within 10 nm, with an analysis of the features of a small area and with high accuracy. To have a deeper understanding of the effect of graphene and inorganic nanoparticles on the surface structure of EP-POSS, XPS was used to analyze the surface elements. The full spectrum is shown in Figure 7. We integrated the peak areas of different elements in the XPS full spectrum of Figure 7 to calculate the surface silicon content. The silicon content data are shown in Table 1. The chemical structure of Si was characterized by the Si spectrum in XPS, and the results are shown in Figure 8. Figure 7 shows that after ZnO/graphene were added, the characteristic peaks of Zn-O and Zn^2+^ appeared at 1022 ev and 1045 ev, respectively, which indicated that there was a small amount of Zn(OH)_2_ in the added nanometer ZnO, and the two were compared in terms of the peak area. The ratio was 2.3:1. After SiO_2_/graphene was added, the surface Si content increases, and the Si content was 7.01%, but compared to the addition of the same amount of graphene, when SiO_2_ was not added, the Si content was 7.19% higher, and the Si-C structure content is higher at 6.78%. This result confirms that the addition of graphene can induce the enrichment of Si on the surface of EP-POSS, and that the Si-C structure bond length is smaller than the C-C bond, which has a stronger shielding effect on the main chain and improves hydrophobicity due to the inertness of the surface of graphene, poor compatibility in EP-POSS, and easy accumulation on the surface. Part of the EP-POSS will grow on the graphene sheet structure and weaken the epoxy segment coat on the POSS structure, so the surface Si content increases.

To verify the effect of the introduction of nanoparticles on the mechanical properties of the coating, EP-POSS nanocomposites were tested for adhesion and wear resistance. The test results are shown in Table 2. According to the test data, the adhesion force of EP-POSS/graphene was 2.10 N/mm^2^, which satisfies the application requirements of the coating. The wear resistance test shows that the friction volume loss was only 1.46 mm^3^, which reached the T level. After SiO_2_ and ZnO nanoparticles were added, the adhesion and wear resistance of EP-POSS nanocomposites still satisfied the application requirements, which shows that the addition of graphene, SiO_2_, and ZnO does not affect the performance of the coating.

## 4. Conclusions

Graphene has a low compatibility with EP-POSS, but the addition of inorganic nanoparticles can improve the compatibility of EP-POSS and graphene. This is because inorganic nanoparticles can be inserted into the sheet structure of graphene to reduce the interlayer interaction, and strengthen the compatibility between graphene and EP-POSS. The compatibility mechanism model is proposed. In addition, the introduction of graphene increases the content of Si-C bonds with high bonding energy, reduces the shielding effect on the cage POSS structure, which is beneficial to the improvement of hydrophobicity. However, graphene forms a stacked structure on the surface of EP-POSS, resulting in an increase in surface roughness, so the contact angle of EP-POSS does not change significantly after adding graphene. Then, SiO_2_, ZnO, and graphene inorganic nanoparticles were introduced into EP-POSS, the contact angle of EP-POSS nanocomposite was greatly reduced due to the hydrophilicity of the nanoparticles, which is not conducive to the hydrophobic properties of the EP-POSS coating, but the adhesion and wear resistance properties of the EP-POSS nanocomposite film meet the actual use requirements.

## Figures and Tables

**Figure 1 nanomaterials-11-00841-f001:**
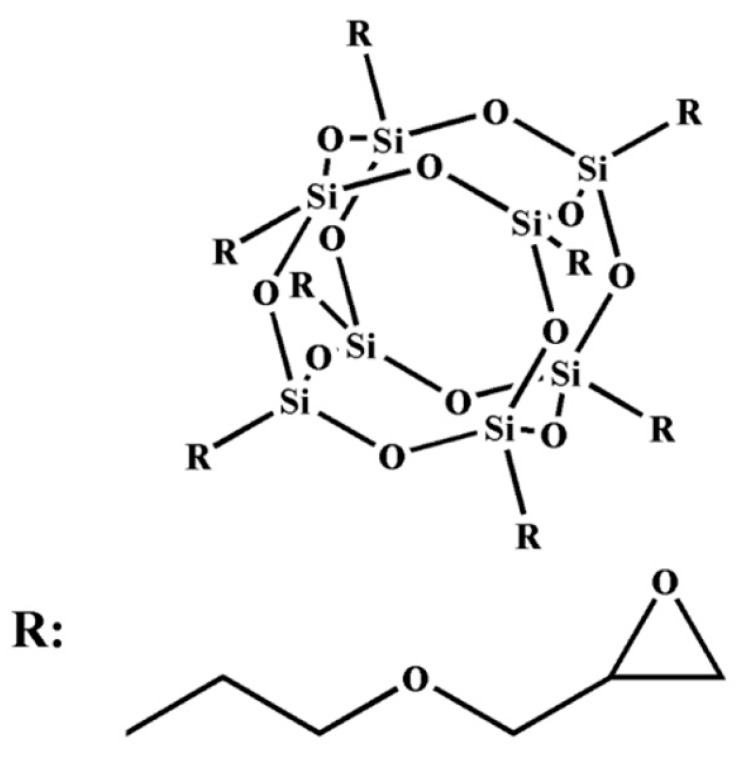
The molecular structure of epoxy resin–polyhedral oligomeric silsesquioxane (EP-POSS).

**Figure 2 nanomaterials-11-00841-f002:**
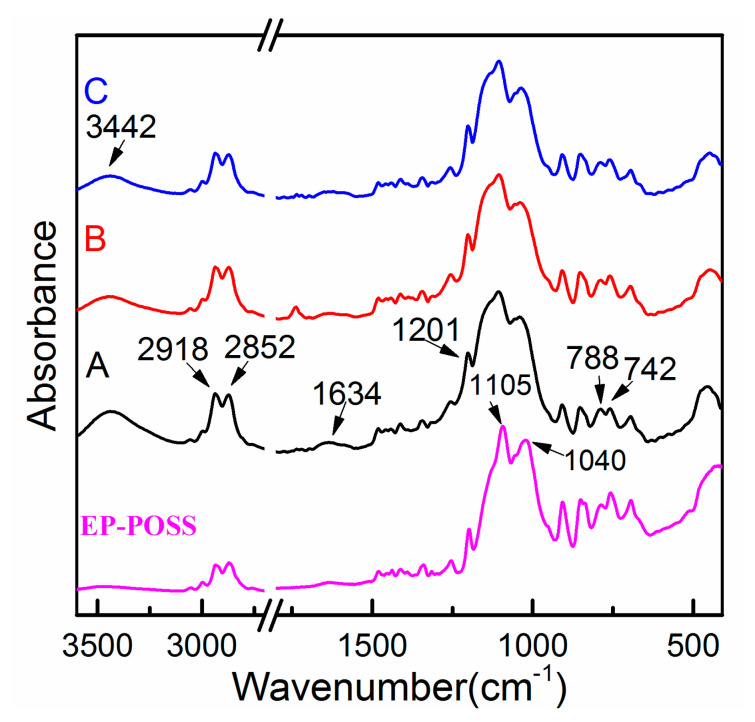
Infrared spectrum of EP-POSS film modified by nanoparticles. A-EP-POSS/SiO_2_/graphene, B-EP-POSS/ZnO/graphene, C-EP-POSS/graphene.

**Figure 3 nanomaterials-11-00841-f003:**
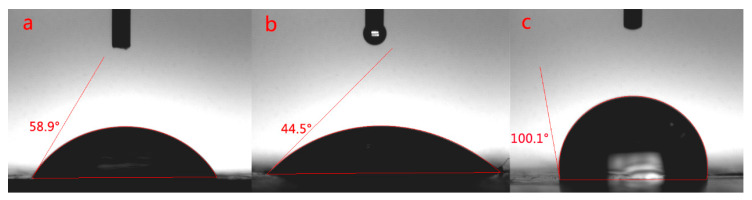
Contact angle test of EP-POSS film modified by nanoparticles. (**a**) EP-POSS/SiO_2_/graphene, (**b**) EP-POSS/ZnO/graphene, (**c**) EP-POSS/graphene.

**Figure 4 nanomaterials-11-00841-f004:**
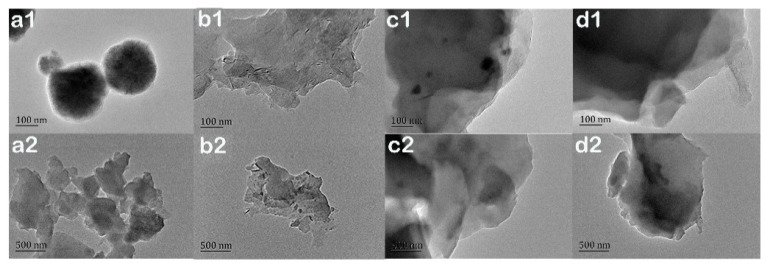
TEM images of EP-POSS nanocomposite film. (**a1**,**a2**) EP-POSS, (**b1**,**b2**) EP-POSS/SiO_2_/graphene, (**c1**,**c2**) EP-POSS/ZnO/graphene, (**d1**,**d2**) EP-POSS/graphene.

**Figure 5 nanomaterials-11-00841-f005:**
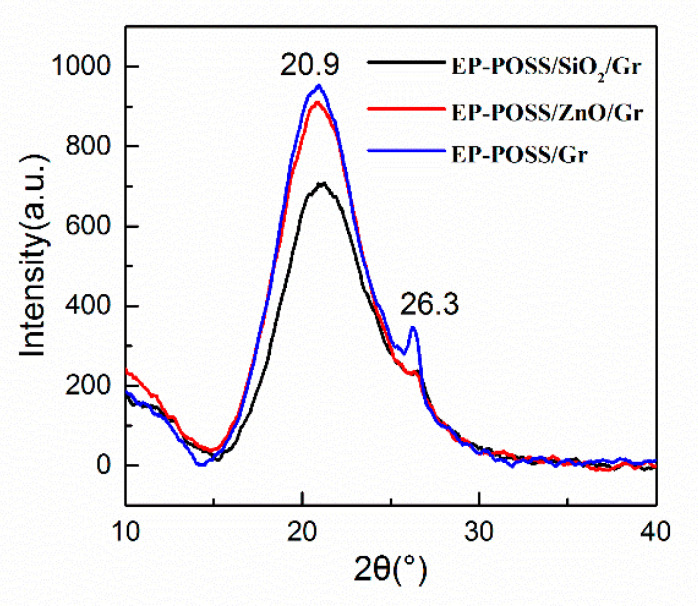
XRD spectra of EP-POSS film modified by nanoparticles.

**Figure 6 nanomaterials-11-00841-f006:**
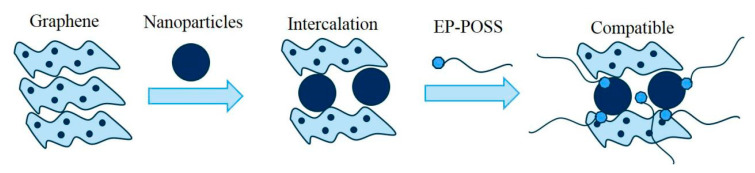
Compatibility model of the EP-POSS/graphene and nanoparticles.

**Figure 7 nanomaterials-11-00841-f007:**
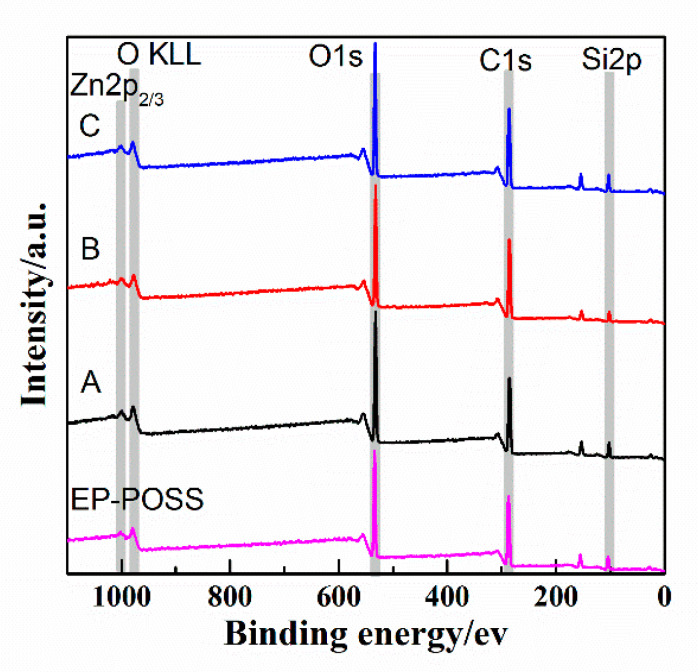
XPS spectrum of EP-POSS film with inorganic nanoparticles. A—EP-POSS/SiO_2_/graphene, B—EP-POSS/ZnO/graphene, C—EP-POSS/graphene.

**Figure 8 nanomaterials-11-00841-f008:**
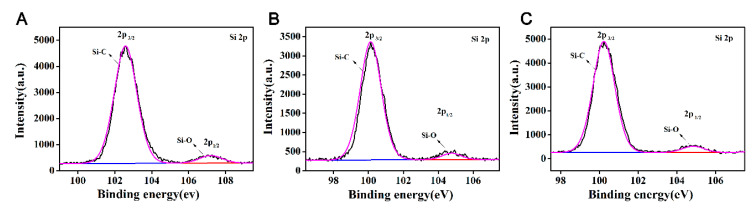
XPS silicon spectrum of EP-POSS with inorganic nanoparticles. (**A**) EP-POSS/SiO_2_/graphene, (**B**) EP-POSS/ZnO/graphene, (**C**) EP-POSS/graphene.

**Table 1 nanomaterials-11-00841-t001:** Surface silicon content of EP-POSS after adding inorganic nanoparticles. (**A**) EP-POSS/SiO_2_/graphene, (**B**) EP-POSS/ZnO/graphene, (**C**) EP-POSS/graphene.

Element	A	B	C
Si/%	7.01	3.52	7.19
Si-C/%	6.48	3.32	6.78
Si-O/%	0.53	0.20	0.41

**Table 2 nanomaterials-11-00841-t002:** Adhesion and wear resistance test of EP-POSS nanocomposite.

Sample	Adhesion (N/mm^2^)	Volume Loss (mm^3^)
EP-POSS/graphene	2.10	1.46
EP-POSS/SiO_2_/graphene	2.18	1.42
EP-POSS/ZnO/graphene	2.12	1.64

## Data Availability

The data presented in this study are available on request from the corresponding author. The data are not publicly available due to business privacy restrictions.

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
