# Peer review of "The Effect of Graphene Nanofiller on the Surface Structure and Performance of Epoxy Resin–Polyhedral Oligomeric Silsesquioxane (EP-POSS)"

_nanomaterials, 2021, doi:10.3390/nano11040841_

Round 1

Reviewer 1 Report

The paper: “ The effect of graphene nanofiller on the surface structure and 2 performance of EP-POSS” by  Yanhong Fang, Ping Wang, Lifang Sun and Linhong Wang describes materials prepared from octakis(glycidyldimethylsiloxy)octasilsesquioxane graphene and nanoparticles of ZnO or SiO2. Several properties of these materials are analyzed and disclosed.

However, there are some suggestions to be revised and some changes to be applied.

  1. The idea of using graphene for the modification of epoxy resin may be very interesting. However, the abstract is presented as the molecule of EP-POSS was mechanically mixed with the other reagents and form the resulting material….. this should be presented in a more precise way. Please change the abstract.
  2. Does the EP-POSS describe the molecule octakis(glycidyldimethylsiloxy)octasilsesquioxane, as presented in Fig.1 or the epoxy resin? The authors use the phrase EP-POSS in a different context. It may be misleading. It is strongly recommended to be specified and changed within the text. As an example – the caption of Fig. 2 -is this the epoxy resin after the curing or the compound EP-POSS?
  3. Is there any covalent bonding created between the EP-POSS resin and the additional components? What is the nature of the binding force?
  4. All the analysis data obtained for samples prepared with ZnO, SiO2 and graphene with EP-POSS resin should be compiled/compared with the same results obtained for pure/neat EP-POSS resin, e.g. in Fig. 3, Table 3 for adhesion Therefore the reader could have the information of real the impact of the additional components on the data obtained.
  5. The information of the % weight content of the prepared sample should be added and compared with the studies presented in the literature.
  6. 2 – a thorough description of the FTIR spectra should be presented. In my opinion, the peak at 788 cm-1 is not strengthened for the sample containing SiO2. The EP_POSS resin contains the Si-O-Si in the majority so the small addition of SiO2 nanoparticles does not change much.
  7. 2 – the should be information added to prove the complete conversion of octakis(glycidyldimethylsiloxy)octasilsesquioxane while the curing process or if there is anything unreacted left.
  8. The conclusions should be verified after all data for EP-POSS resin are presented and corresponded with the examined samples and finally compared with the similar studies presented in the literature.

The editorial part:

  • Figure 2 and others – the caption should be added as a caption, not as a text of the manuscript
  • Figure 4 – please change the font color on the SEM images to improve their readability
  • Figure 8 should be enlarged and a respective caption should be added
  • Table 3 – the sample names should be corrected – at the moment there is “grapheme”
  • Please verify the Reference style. For Nanomaterials it should be presented as follows the example of papers:

Nakagaki, S.; Ferreira, G.; Marcalb, A.; Ciuffi, K. Metalloporphyrins Immobilized on Silica and Modified Silica as Catalysts in Heterogeneous Processes. Curr. Org. Synth. 2014, 11, 67–88

  • English should be also checked once more.

Author Response

Dear Reviewer:

Hello, the modified places have been marked in red, and the language grammar and spelling are finally revised by MDPI.Thank you very much for your attention and the referee’s evaluation and comments on this manuscript. We have revised the manuscript according to referee’s detailed suggestions. Revised portion are marked in red. Enclosed please find the responses to the referees. We sincerely hope this manuscript will finally meet the requirement of Nanomaterials. Thank you very much for all your help and looking forward to hearing from you soon.

With kindest regards,Yours sincerely,Ping Wang

Reviewer 2 Report

The manuscript by these authors is an interesting contribution to the surface modification of epoxy- POSS reinforced matrix by adding graphene nanofiller. The performance of the modified EP-POSS was investigated and the results discussed. The work is certainly of impact on the scientific community, due to the importance and the interest in epoxy, POSS and graphene. Anyway, the text needs a revision in terms of grammar and revision of typos. Furthermore, the introduction must be improved by citing relevant references on specific statements addressed by the authors (see attached .pdf). These and other suggestions are reported in the attached .pdf.

Author Response

Dear Reviewer:

Thank you for carefully reviewing this article. We have carefully revised all the questions and errors you raised, including five parts: Abstract, Introduction, Experiment, Results and Conclusion, and all revised paragraphs are marked in red. Please see the attachment. Thanks again for your patience.

Yours sincerely,

Yanhong Fang

Reviewer 3 Report

In this work epoxy-POSS (EP-POSS) has been  modified with graphene, nano-SiO2 and nano- ZnO, and the obtained composites characterized by FTIR, SEM-EDS, XRD and XPS techniques.  The influence of nanoadditives on the properties of EP-POSS has been studied and discussed. Some issues need to be further addressed:

  • Title, Abstract, Keywords: please expand the abbreviation "EP-POSS" that is not commonly used,
  • Introduction:  "Scientists at the Massachusetts Institute of Technology recently discovered that graphene [1] is a carbon material composed of a single-atom structure..." - it has been well-known since two decades, not "recently",
  • Introductions has certain flaws and random paragraphs, please ensure it is coherent,
  • 2.1 - please provide short description on EP-POSS preparation protocol,
  • 3 - "Infrared spectroscopy is a qualitative method to analyze the material structure, and it is commonly used to study the characteristic functional groups of materials" - trivial, please delete, the same for other methods,
  • Fig. 4 - SEM magnification shows micrometric agglomerates, please provide TEM images to show nanoparticles, otherwise the materials are not "nanocomposites",
  • Conclusion needs to be re-written as it does not show all the main achievements of the work,
  • References are prepared very carelessly, e.g. Refs. 1, 3, 18, 20.  Please add: Preparation and properties of POSS/epoxy composites for electronic packaging applications,  Materials & Design, Volume 30, Issue 1, January 2009, Pages 1-8; Nanostructured Epoxy/POSS Composites: Enhanced Materials for High Voltage Insulation Applications, June 2015, IEEE Transactions on Dielectrics and Electrical Insulation 22(3):1594-1604.

Author Response

Dear Reviewer:

Thank you very much for your attention and the referee’s evaluation and comments on this manuscript. We have revised the manuscript according to referee’s detailed suggestions. Revised portion are marked in red. Enclosed please find the responses to the referees. We sincerely hope this manuscript will finally meet the requirement of Nanomaterials. Thank you very much for all your help and looking forward to hearing from you soon.

With kindest regards,Yours sincerely,Ping Wang

Round 2

Reviewer 1 Report

Dear Authors,

  1. Please verify the title, as there is a typo in the name: "Expoy resin-polyhedral oligomeric silsesquioxane" - it should be "Epoxy resin-polyhedral oligomeric silsesquioxane". This should be verified throughout the article, as there are a few places with this mistake.
  2. Could you think over the idea of putting the explanation of the abbreviation EP-POSS in brackets in the title?
  3. Point 2.1 concerning Rwa Materials - the part marked red - is not written in understandable English. It is strongly suggested to be verified.
  4. In your response, you wrote "and we believed that
    Octakis (Glycidyldimethylsiloxy) Octasilsesquioxane had been completely transformed, but the purity was low" - where is the information on this in the revised paper? How should it be verified whether it is completely consumed during the curing process?
  5. Also, the information of FT-IR spectra description is still not complete.
  6. The information of the % weight content of the prepared sample was put in the authors' response file but is not present in the revised version of the manuscript. Please verify. 
  7. The caption of Figure 3, 4 - please format as it should be with a), b) and c) etc.
  8. The conclusions were not changed as it is in the response to the reviewer file. Please change it.

Author Response

Hello, thank you for your support for my article. Please see the attachment.

Reviewer 2 Report

This revised version has been improved with respect to the previous one. Anyway, not all the suggestions were receipt by the authors. Like for instance the caption of figure 4 remains meaningless and needs to be written better. Also in the introduction, I would suggest adding some relevant references on POSSs, like for instance a couple of papers by Dr. Lichtenhan [POSS-Based Polymers. Polymers 201911, 1727. https://doi.org/10.3390/polym11101727; Polyhedral oligomeric silsesquioxanes: building blocks for silsesquioxane-based polymers and hybrid materials.  Comments on Inorganic Chemistry, 1995 - Taylor & Francis https://doi.org/10.1080/02603599508035785

Author Response

(The authors gave the same response as above.)

Reviewer 3 Report

In the revised version authors have not addressed my comment  "Fig. 4 - SEM magnification shows micrometric agglomerates, please provide TEM
images to show nanoparticles, otherwise the materials are not "nanocomposites"".  The reply should include TEM images.  

Author Response

(The authors gave the same response as above.)

Round 3

Reviewer 3 Report

In the revised version of the manuscript authors present Figure 4. "EP-POSS and TEM after adding fillers.".  Apart from weird English (POSS and TEM?),  in the 2.3. Test method section SEM technique is described  "The surface of the sample was sprayed with gold and tested with a JSM-6060LV JEOL scanning electron microscope and an Oxford Instruments’ Inca 350 X-act energy spectrum analysis EDS." and TEM is not mentioned.   Please check it.   It could be either Figure 4. "SEM images of EP-POSS with fillers" or ""TEM images of EP-POSS with fillers" - in the latter case please replace SEM description with TEM one. 
